# Enumeration and Characterization of Circulating Tumor Cells in Patients with Hepatocellular Carcinoma Undergoing Transarterial Chemoembolization

**DOI:** 10.3390/ijms24032558

**Published:** 2023-01-29

**Authors:** María L. Espejo-Cruz, Sandra González-Rubio, Juan J. Espejo, Javier M. Zamora-Olaya, Rafael M. Alejandre-Altamirano, María Prieto-Torre, Clara I. Linares, Marta Guerrero-Misas, Pilar Barrera-Baena, Antonio Poyato-González, Marina Sánchez-Frías, María D. Ayllón, Manuel L. Rodríguez-Perálvarez, Manuel de la Mata, Gustavo Ferrín

**Affiliations:** 1Maimonides Biomedical Research Institute of Córdoba (IMIBIC), University of Córdoba, 14004 Cordoba, Spain; 2Biomedical Research Network Center for Liver and Digestive Diseases (CIBERehd), 28029 Madrid, Spain; 3Department of Interventional Radiology, Reina Sofia University Hospital, 14004 Cordoba, Spain; 4Department of Hepatology and Liver Transplantation, Reina Sofia University Hospital, 14004 Cordoba, Spain; 5Department of Pathology, Reina Sofia University Hospital, 14004 Cordoba, Spain; 6Department of Hepatobiliary Surgery and Liver Transplantation, Reina Sofia University Hospital, 14004 Cordoba, Spain

**Keywords:** circulating tumor cells, circulating cancer stem cells, liquid biopsy, hepatocellular carcinoma, transarterial chemoembolization, spheroids

## Abstract

Circulating tumor cells (CTCs), and particularly circulating cancer stem cells (cCSC), are prognostic biomarkers for different malignancies and may be detected using liquid biopsies. The ex vivo culture of cCSCs would provide valuable information regarding biological aggressiveness and would allow monitoring the adaptive changes acquired by the tumor in real time. In this prospective pilot study, we analyzed the presence of EpCAM^+^ CTCs using the IsoFlux system in the peripheral blood of 37 patients with hepatocellular carcinoma undergoing transarterial chemoembolization (TACE). The average patient age was 63.5 ± 7.9 years and 91.9% of the patients were men. All patients had detectable CTCs at baseline and 20 patients (54.1%) showed CTC aggregates or clusters in their peripheral blood. The increased total tumor diameter (OR: 2.5 (95% CI: 1.3–4.8), *p* = 0.006) and the absence of clusters of CTCs at baseline (OR: 0.2 (95% CI: 0.0–1.0), *p* = 0.049) were independent predictors of a diminished response to TACE. Culture of cCSC was successful in five out of thirty-three patients, mostly using negative enrichment of CD45^−^ cells, ultra-low adherence, high glucose, and a short period of hypoxia followed by normoxia. In conclusion, the identification of clusters of CTCs before TACE and the implementation of standardized approaches for cCSC culture could aid to predict outcomes and to define the optimal adjuvant therapeutic strategy for a true personalized medicine in hepatocellular carcinoma.

## 1. Introduction

The enumeration of circulating tumor cells (CTCs) has emerged as a useful strategy for cancer surveillance. However, CTCs comprise a heterogeneous population of cancer cells with a wide range of invasiveness and metastatic potentials, which are associated with therapeutic response and clinical outcomes [1]. The isolation approach is not standardized and different methodologies for characterization of CTCs may result in varying findings. In hepatocellular carcinoma (HCC), CTCs can be identified using liver and HCC specific markers (e.g., glypican-3 and asialoglycoprotein receptor), epithelial markers (e.g., epithelial cell adhesion molecule -EpCAM- and cytokeratin -CK-), epithelial-to-mesenchymal transition (EMT) markers (e.g., vimentin and twist), and stem cell markers (e.g., CD44 and CD90). The best approach would probably result from combining several of the above markers.

The EMT process is characterized by a loss of the expression of epithelial markers, which confer tumor cells with increased invasiveness, intravasation capacity, and/or drug resistance, ultimately facilitating the spread of cancer. Thus, the use of epithelial markers alone to identify CTCs could entail the loss of CTC subpopulations with increased aggressiveness, such as circulating cancer stem cells (cCSC). EpCAM is a typical surface marker of HCC cells with stem cell features [2,3] and EpCAM^+^ CTCs derived from HCC patients display high tumorigenic potential in vivo [4]. Accordingly, epithelial markers for CTC isolation and/or enumeration have been associated with clinicopathological characteristics of cancer [5].

The ex vivo culture of CTCs is appealing, as it could allow testing tumor sensitivity to different chemotherapeutics. In addition, given the non-invasive nature of this analysis, CTC culture could be repeated over time, allowing monitoring of the acquisition of new mutations contributing to drug resistance, thus anticipating tumor progression and allowing early access for second line therapies [6,7]. However, CTC culture is challenging. The main shortcoming is the small number of CTCs in the bloodstream which makes isolation and enrichment paramount pre-requisites, with different approaches available [8]. Among the methods based on positive selection of CTCs [9,10], the IsoFlux^TM^ system displays a great sensitivity in the identification of CTCs from HCC patients [11]. However, positive selection methods rely on the cellular expression of specific surface markers and could be missing small subpopulations of CTCs with high tumorigenicity. To overcome this caveat, negative selection methods based on specific physical or biochemical properties of CTCs, including cellular filtration, fractionation, or depletion, have been used [7,12,13]. Even if successfully isolated, the ex vivo culture of CTCs is far from standardized and there is an ongoing debate regarding the optimal scaffold, physical conditions, nutrients, and supplements [5].

Transarterial chemoembolization (TACE) is the standard of care for patients with intermediate stage HCC or for patients at earlier stages who are not eligible for surgical resection or liver transplantation [14]. TACE is a minimally invasive radiological procedure during which a chemotherapeutic agent is directly infused into the main arterial supplier of the tumor, followed by the occlusion of the involved vessel with 100–500 micron-sized embolic particles. In recent years, promising new technological advances have been developed, aimed at improving the stability of pharmaceutical agents and the clinical performance of TACE [15,16]. Currently, radiological response one month after the procedure may be achieved in the majority of patients, but tumor recurrence is almost universal, conditioning a modest survival benefit ranging from 20 to 36 months [17]. Since TACE induces a dual tumoricidal effect driven by hypoxemia and chemotherapy, prior HCC cell isolation and culture would offer the opportunity to simulate the effect of TACE on each particular tumor in order to personalize the radiological procedure, to select candidates for adjuvant systemic therapies, or even to consider transarterial radioembolization over TACE in those tumors with chemo-resistant CTCs.

In this pilot study, we aimed to explore the feasibility of CTC enumeration, isolation, and culture in patients with HCC undergoing TACE as a first line therapy. In addition, we aimed to analyze the influence of baseline enumeration of CTCs on tumor features and outcomes after TACE.

## 2. Results

### 2.1. Clinicopathological Characteristics of the Study Population

In all, 37 HCC patients were prospectively enrolled. Table 1 shows the clinicopathological characteristics of the included cohort. The average age was 63.5 ± 7.9 years, and 91.9% (n = 34) were men. Alcoholic liver disease was the main etiology (73.0%; n = 27), followed by chronic hepatitis C (37.8%; n = 14), chronic hepatitis B (13.5%; n = 5), and non-alcoholic fatty liver disease (10.8%; n = 4). The median serum concentration of alpha-fetoprotein (AFP) before TACE was 10.3 ng/mL (interquartile range (IQR) 3.7–49.0). Most patients had Child–Pugh class A cirrhosis (86.5%; n = 32). The MELD score was 9.0 (IQR 8.0–11.0). A single nodule HCC was observed in 48.6% (n = 18) of patients. Patients with multinodular disease had two nodules (29.7%; n = 11), three nodules (8.1%; n = 3), or four nodules (13.5%; n = 5). The total tumor diameter (i.e., the sum of the diameter of all nodules) was 3.8 cm (IQR 2.9–5.7).

### 2.2. Relationship between the Clinicopathological Features of the Patients and the CTC Count

All patients included in this study had detectable CD45^−^ CK^+^ CTCs and double positive CD45^+^ CK^+^ cells immediately before TACE. The median counts of CTCs and double positive CD45^+^ CK^+^ cells were 34.0 (IQR 21.5–100.0) and 28.0 (IQR 16.0–59.0), respectively. Combining CTCs and double positive cells, the median cell count was 68.0 cells (IQR 41.0–138.5). The relationship between clinicopathological features and CTC count is shown in Table 2. Hypervascular tumors showed increased number of CTCs (*p* = 0.026). The diameter of the main nodule < 3 cm was associated with increased double positive CD45^+^ CK^+^ cells (*p* = 0.024). CTC aggregates or clusters were detected in 54.1% of patients (n = 20). These patients showed an increased CTC count (23.0 (IQR 14.0–42.5) vs. 72.5 (IQR 33.3–191.0), *p* = 0.001), but had a similar double-positive cell enumeration (23.0 (IQR 14.0–40.5) vs. 35.0 (IQR 19.8–97.0), *p* = 0.057) (Appendix A). CTC count did not correlate with serum AFP (*r* = −0.184, *p* = 0.290), aspartate aminotransferase (*r* = −0.083, *p* = 0.623), alanine aminotransferase (*r* = 0.124, *p* = 0.464), or gamma-glutamyl transferase (*r* = 0.096, *p* = 0.570).

### 2.3. Prognostic Value of CTC Count in HCC Patients Undergoing TACE

The relationship between baseline CTC counts and radiological response one month after TACE is shown in Table 3. Among the 37 patients receiving TACE, 45.9% of patients (n = 17) did not achieve complete radiological response one month after the procedure; partial response was observed in 27.0% of patients (n = 10), stable disease occurred in 10.5% (n = 4), and disease progression was shown in 8.1% of patients (n = 3). Advanced BCLC stage (OR: 5.0 (95% CI: 1.1–23.8), *p* = 0.041), multinodular disease (OR: 4.5 (95% CI: 1.1–17.9), *p* = 0.035), larger tumor diameter (OR: 2.2 (95% CI: 1.3–3.9), *p* = 0.005), targeting more than one nodule during TACE (OR: 4.3 (95% CI: 1.1–17.4), *p* = 0.041), and absence of CTC clusters at baseline (OR: 0.2 (95% CI: 0.1–0.9), *p* = 0.039) were associated with poor response to TACE. CTC count at baseline had no influence on the therapeutic response.

In the multivariate analysis, an increased total tumor diameter (OR: 2.5 (95% CI: 1.3–4.8), *p* = 0.006) and the absence of CTC clusters (OR: 0.2 (95% CI: 0.0–1.0), *p* = 0.049) were independent predictors of a poor response to TACE. The multivariate model had an area under the receiver operating characteristic curve (AUROC) of 0.90 (95% CI: 0.7–1.0), with a sensitivity of 81.8% (95% CI: 63.4–100.0), a specificity of 86.7% (95% CI: 66.1–100.0), a positive predictive value of 90.0% (95% CI: 74.4–100.0), and a negative predictive value of 76.5% (95% CI: 53.4–99.6) (Figure 1 and Table 4).

The median follow-up after TACE was 12.0 months (IQR 6.0–20.0). Patients who underwent liver transplantation during follow-up (n = 11, 29.7%) were censored in the survival analysis. There were twelve deaths (32.4%), eight of them (21.6%) were related with tumor progression. Neither the CTC count nor the presence of CTC clusters were associated with transplant-free survival rates (*p* = 0.881 and *p* = 0.796, respectively; Figure 2 and Table 5).

### 2.4. Ex Vivo Culture of CTCs

#### 2.4.1. Culture of EpCAM^+^ CTCs Isolated by the IsoFlux System

First, we tested the IsoFlux system to isolate viable EpCAM^+^ CSC from established HCC cell lines. After obtaining cellular spheres from cell lines HepG2 and Huh7, they were disaggregated with trypsin and immunomagnetically captured with the IsoFlux system, following the same protocol used for CTC enumeration in our cohort. As shown in Figure 3, a high number of these cells were CK^+^ CD45^−^ when fixed and stained for immunofluorescence, with a reduced proportion of double positive CK^+^ CD45^+^ cells (Figure 3A). Retrieved cells bound to the magnetic microbeads were immediately placed in culture under normoxia and ultra-low adherence (ULA) conditions to favor the growth of CSC spheres. Using this approach, isolated cells were able to generate spheres after 2 weeks (Figure 3B). Furthermore, these cells retained this ability when they were enzymatically disaggregated and cultured again under the same conditions.

Next, we reproduced the same approach to obtain CSC spheres from peripheral blood samples of HCC patients undergoing TACE in our study. We failed to obtain CSC-like spheres when we used the IsoFlux system to isolate EpCAM^+^ CTCs from the first 11 patients included in the study. In most of the samples, we only observed single cells and clusters of cells surrounded by magnetic microbeads, which were dragged during cell isolation. However, cell growth was not observed.

#### 2.4.2. Culture of CD45^−^ CTCs Isolated by Negative Enrichment

After the negative results obtained with the IsoFlux system, we considered negative enrichment of CD45^−^ cells by immunocapture and subsequent filtration on magnetic columns. This method allowed to obtain cCSC-like spheroids in five of the twenty-two HCC patients analyzed when CD45^−^ cells were cultured in the same conditions (high glucose, normoxia, and ULA). A complete radiological response after TACE was observed in 80% of these patients (4 out of 5) (Table 6).

Initially, cell cultures were maintained under standard normoxic conditions. The spheroids showed discrete growth that increased with time at different rates. In some patients, the spheroids began to develop elongations after a few days in culture (Figure 4, black arrows). These structures were capable of connecting different spheroids to each other and even anchoring the spheroid to the substrate; this phenomenon had not been previously observed in our experiments using HepG2 and Huh7 cell lines. The ex vivo culture of the spheroids did not last longer than 10 days in any case. Thereafter, the spheroids stopped growing and the cells rapidly died.

From patient 24 onwards, we decided to maintain cell cultures in a hypoxia incubator chamber to study the effect of a deprived oxygen supply on the growth capacity of CTCs. We previously found that this condition favored the growth of cell spheres in the HepG2 cell line. In contrast, glucose depletion negatively affected the generation and growth of CSC spheroids. Thus, despite the fact that CSC spheroids are difficult to quantify and measure when they are grown under ULA conditions, mainly because they usually merge in the center of the well of the culture plate causing larger conglomerates, the described effects on cell growth were evident to the naked eye (Figure 5). Therefore, we concluded that hypoxia for 3 days could be effective in stimulating the growth of cCSC spheroids from HCC patients.

In order to compare the effect of oxygen supply on the growth of spheroids from CTCs, 10 samples of enriched CD45^−^ cells were maintained under normoxia and hypoxia. We observed spheroids in both culture conditions in two of these samples (Table 6). As we observed for the HepG2 cell line, hypoxia stimulated the growth of cCSC spheres (Figure 6). Despite this, the spheroids started to die off rapidly after 7–10 days of ex vivo culture.

## 3. Discussion

CTCs may facilitate cancer progression and resistance to therapy, thus resulting in poor prognosis. CTCs, and particularly cCSC, are hard to identify because of their scarcity compared with other blood cell populations. In the present prospective pilot study, baseline enumeration of CTCs did not influence outcomes in patients with HCC undergoing TACE. However, the identification of clusters of CTCs before TACE was associated with improved radiological response irrespective of the size of the tumor.

Despite the difficulties for establishing cell lines from CTCs [18], short-term cultures of cCSC could provide valuable information. The use of magnetic beads coated with anti-EpCAM for CTC isolation with subsequent culture under ultra-low adherence did not allow obtaining stable CTC spheres from peripheral blood of patients with HCC. In contrast, under similar culture parameters, we obtained spheres from established HCC cell lines, suggesting that microbeads could be hindering the growth of the CTC spheres. Novel methodologies allowing self-release and automatic degradation of enrichment beads under an acidic environment during cell culture could improve cell viability [19]. A negative enrichment strategy based on depletion of CD45^+^ cells was more efficient to obtain cCSC-like spheroids, but still suboptimal (5 out 22 patients). Obtaining a stable culture of CTCs was not possible as CTC spheroids died rapidly, precluding a sufficient cell mass for further marker expression analyses. It could be possible that, in addition to the cell enrichment methodology used, CTCs from different HCC patients require different culture conditions to form spheres in suspension, as previously reported in other cancer cell lines [20].

An interesting finding in CTC cultures was the long structures acquired by cCSC spheroids, which allowed them to contact each other and adhere to the substrate. ROCK inhibitors are frequently used for maintaining (cancer) stem cells since they can block apoptosis and increase survival and cloning efficiency without affecting pluripotency [21,22]. Despite that fact that we did not observe morphological changes when Y27632 was used for the generation and maintenance of CSC spheroids from HCC cell lines, ROCK inhibition by Y27632 can promote the formation of microtubule-based structures which increase the reattachment efficacy of suspended breast cancer cells and could enhance the metastatic potential of non-adherent CTCs [23]. In contrast, Y-27632 also inhibits cellular migration and morphological change of rat ascites hepatoma (MM1) cells [24]. In mesenchymal stromal cell/stem cells [25] and cell line PC12, Y27632 induces morphological changes in neuron-like cells by a mechanism that could involve pAKT [26]. In addition, nerves are important components of the cancer microenvironment and may have a significant role in cancer progression [27]. CSC could be a source for building a neural network within tumor tissues in vivo [28]. If cCSC from liver cancer patients could differentiate into neural cells in our culture conditions, it could be useful for the understanding of molecular mechanisms underlying perineural invasion in cancer and its relationship with metastatic spread [29].

TACE therapy is associated with hypoxia and nutrient deprivation, but still, between 27% and 72% of HCC patients do not obtain a complete tumor necrosis after TACE [30]. It may well be that sub-lethal hypoxia and low glucose could even favor tumor recurrence and progression [31]. Hypoxia, but not glucose depletion, promoted the growth of CSC spheroids from established HCC cell lines. Similarly, hypoxia seemed to stimulate ex vivo proliferation of spheroids from CD45^−^-enriched CTCs from two HCC patients. However, spheroids did not survive long in culture. New methodologies are needed for stabilizing ex vivo cultures of cCSC in order to elucidate the biological nature of these cancer-initiating cells.

CTC enumeration is a useful prognostic tool in cancer patients. Vascular invasion, high serum AFP, or an advanced BCLC stage are clinicopathological features of HCC patients and have been associated with EpCAM^+^ CTCs [32,33]. The CTC count can predict outcomes of HCC patients undergoing potentially curative therapies such as liver transplantation, liver resection, radiotherapy, or TACE [5]. Previous studies evaluating the prognostic value of CTC count before TACE therapy showed contradictory results. The negative enrichment of CD45^−^ cells combined with a quantitative real-time PCR-based platform showed that EpCAM mRNA^+^ in HCC patients (n = 56) before TACE is associated with a higher risk of tumor progression [34]. These results were consistent with those obtained with the CellSearch system in a cohort of 89 patients with HCC undergoing TACE [35]. Using negative enrichment and CTC identification by iFISH (CEP8/CD45/DAPI markers) in a larger cohort (n = 155), Xiaoxia Wu et al. associated positive serum CTC counts before TACE with a shorter overall survival and disease-free survival. Again, high CTCs before treatment correlated with a poor prognosis [35,36]. Using this methodology in a sample of 43 HCC patients, the number of CD45^−^ CEP8^+^ CTCs before TACE was also identified as an independent biomarker for predicting overall survival of patients with advanced HCC [37]. In contrast, positive enrichment of EpCAM^+^ CTCs with magnetic beads and columns, and subsequent counting of nucleated CK^+^ CD45^−^ cells, did not show any prognostic value in 42 HCC patients undergoing TACE [38]. Similarly, when CTCs were identified by flow cytometry as CD45^−^ and positive for markers ASGPR, CD146, and PD-L1, the CTC count before TACE did not influence the tumor recurrence rate and overall survival [39]. In the present study, using the highly sensitive IsoFlux system for the identification of CTCs in HCC [11], we found no association between the CD45^−^ CK^+^ cell count at baseline and TACE response. An analysis combining both CD45^−^ CK^+^ cells and CD45^+^ CK^+^ cells, since the latter could be CTCs in cancer patients [40], showed similar results. Some studies have suggested that a significant fraction of CD45^+^ CK^+^ cells also express CD68, which is a marker associated with tumor-associated macrophages [41]. In addition, we also observed double positive cells in CSC derived from established cell lines of HCC. Thus, although the role of CD45^+^ CK^+^ cells has not yet been defined, additional staining for CD45 might not be specific of hematopoietic cells. The optimal surface marker or, more likely, a combination of surface markers, able to unequivocally identify CTCs with metastatic potential is still to be determined.

TACE may be more effective for HCC tumors with a rich blood supply because the chemotherapeutic agent would diffuse more efficiently within the tumor and the subsequent vascular occlusion would provoke a more intense ischemic damage [42,43]. In the present study, hypervascular tumors were associated with a higher CTC count, which in turn was related to the presence of CTC clusters. Moreover, patients who exhibited a complete radiological response to TACE were those showing CTC clusters at baseline, irrespective of the tumor burden. This is an unexpected and intriguing finding. The angiogenesis promoter VEGF enhances spontaneous metastasis by inducing intravasation of heterogeneous tumor cell clusters [44], with the VEGF-VEGFR signal blocking being a potential target to reduce trans-endothelial migration, CTC colony number, and metastases [45]. It may well be that clusters of CTCs accessing the bloodstream through the profuse HCC vascularity attract the attention of the immune system more effectively than isolated CTCs. In such a case, the immune system could use these CTC clusters to recognize tumor-derived antigens and to orchestrate a more powerful response against the primary tumor, thus explaining in part the relationship between peri-tumor inflammatory response and better outcomes in HCC [46].

The present study has inherent limitations. As a pilot study, we could not estimate sample size and the reduced number of patients included may have weakened some analyses. In addition, the evolution of CTC enumeration and clusters after TACE has not been assessed and could provide relevant information to predict outcomes or to determine the optimal timing to initiate systemic therapies.

## 4. Materials and Methods

### 4.1. Study Design, Population, and TACE Procedure

This is a prospective pilot study including a consecutive cohort of adult patients diagnosed with liver cirrhosis and early or intermediate stage HCC (according to Barcelona clinic liver cancer (BCLC) classification), who underwent TACE as a first line therapy from January 2019 to June 2022. Patients were followed after their procedure until death or November 2022. Patients undergoing liver transplantation were censored for survival analysis. Exclusion criteria were portal vein thrombosis, severe impairment of liver function (Child–Pugh class C), moderate–severe ascites, prior therapies for HCC, and being admitted in the waiting list for liver transplantation at baseline.

Patients were hospitalized 24 h before TACE according to local clinical practice to verify the absence of analytical contraindications. Under conscious sedation using intravenous midazolam, fentanyl, and local anesthesia, a 4–6 Fr vascular sheath was inserted through the femoral artery or radial artery at the discretion of the interventional radiologist. The vascular sheath was placed in the mesenteric artery to perform an indirect portography, which allowed to ensure portal vein patency. Afterwards, the sheath was advanced through the intrahepatic arterial branches to locate the typical aberrant tributaries of the tumor. Then, 100–500 micron-sized drug-eluting beads with doxorubicin were infused in the main arterial supplier of the tumor [47]. The process was repeated in each tumor location with a supra-selective approach. After the procedure, patients remained hospitalized for 24 h and were discharged afterwards in the absence of complications. The radiological response to TACE was assessed either by dynamic computed tomography or magnetic resonance one month after the procedure using the modified response evaluation criteria in solid tumors (mRECIST) [48], which assess changes in tumor size and vascularity compared to baseline, classifying patients into the following categories: complete response, partial response, stable disease, and progressive disease. For the present study, patients showing a complete response were classified as responders and the remaining categories were classified as non-responders. Clinical follow-up visits were scheduled at 3, 6, 9, and 12 months after TACE and then every 6 months until the end of the study. Liver function, kidney function, and serum AFP levels were determined at each visit. Patients underwent dynamic imaging techniques every 3 to 6 months to assess tumor recurrence/progression in accordance with our usual clinical practice.

All patients were required to sign an informed consent document in order to participate in the study. The research study was conducted according to the Declaration of Helsinki and the study protocol was approved by the Andalusian Research Ethics Committee (code 0117-N-18) as part of a research project (code PI18/01736).

### 4.2. Evaluation of Tumor Vascularity

Tumor vascularity was assessed by the enhancement effect on the arterial phase of angiography during the TACE procedure by the interventional radiologist, as described in [49]. Easily visible tumors by celiac and selective angiography were classified as hypervascular lesions, while tumors minimally visible by celiac angiography but evident by selective angiography were assessed as mildly vascular. Finally, those tumors described in the computed tomography or magnetic resonance at baseline, which were not detected by celiac angiography or selective angiography, were assessed as hypovascular.

### 4.3. Sample Collection and Data Management

Peripheral venous blood samples were obtained by phlebotomy within 24 h prior to TACE. Samples were collected in a Vacutainer tube (K2-EDTA) (BD Bioscience, Franklin Lakes, NJ, USA) and kept at room temperature (RT) until analysis (less than 24 h). Clinical characteristics and outcomes were recorded in a dedicated electronic datasheet, which was anonymized. Any missing value was retrieved from the patient electronical records. The investigators evaluating CTC enumeration and performing cell cultures were blinded to the patient clinical outcomes.

### 4.4. Cell Culture

Human HCC cell lines HepG2 and Huh7 were acquired from the European Collection of Authenticated Cell Cultures (ECACC) and maintained under adherent conditions in high glucose (4.5 g/L) Dulbecco’s Modified Eagle’s Medium (DMEM) (Thermo Fisher Scientific, Waltham, MA, USA), which contained 0.584 g/L L-glutamine and 1 mM sodium pyruvate and was supplemented with 1× antibiotic-antimycotic and 10% FBS (Biowest, Nuaillé, France). Cells were incubated in a humidified atmosphere (21% O_2_) at 37 °C and 5% CO_2_.

To promote the growth of CSCs derived from both HCC cell lines or CTCs, cells were maintained in 6-well or 24-well ULA plates (Corning, Kennebunk, ME, USA) under normoxia (21% O_2_) or hypoxia (1% O_2_) at 37 °C and 5% CO_2_. The culture medium consisted of high-glucose DMEM/F12 and/or glucose-depleted SILAC DMEM/F12, in either case supplemented with 20 ng/mL recombinant human bFGF, 20 ng/mL recombinant human EGF, 1× of supplements N2, B27, and insulin-transferrin-selenium (Thermo Fisher Scientific), 0.01 mM ROCK inhibitor Y27632 (Calbiochem, Darmstadt, Germany), and 1× antibiotic-antimycotic (Biowest). Additionally, the SILAC DMEM/F12 medium was also supplemented with 1× GlutaMAX (Thermo Fisher Scientific), 147.5 mg/L L-arginine, 91.25 mg/L L-lisine (Sigma-Aldrich, St. Louis, MO, USA), and 1 g/L D-glucose (Sigma-Aldrich).

To obtain CSC spheroids from established cell lines, we seeded 5000 cells in a 6-well ULA plate containing 2 mL of DMEM/F12 culture medium supplemented as described above. The medium was supplemented every 48 h and, at the time the spheroids reached a size between 150 and 200 µm, cells were trypsinized and used for experiments.

The CTC culture was maintained for up to 15 days, assessing the presence of CSC-like spheroids every other day with a Nikon Eclipse Ti-S inverted microscope (Nikon, Tokyo, Japan).

### 4.5. CTC Isolation from Peripheral Blood

CTC isolation was independently performed for enumeration and for culture. For cell counting, CTCs were isolated by positive enrichment using the IsoFlux system (Fluxion Biosciences, Oakland, CA, USA) within 24 h of sample extraction. For cell culture, to increase cell viability, CTCs were isolated immediately after the sample extraction, using strategies of positive or negative enrichment under controlled conditions of sterility and constant room temperature. In any case, density gradient centrifugation with Ficoll (GE Healthcare, Chicago, IL, USA) was used in the first step with Leucosep tubes (Greiner Bio-One, Kremsmünster, Austria), which guaranteed the reproducibility of the cell isolation technique. Briefly, 7 mL of peripheral blood was centrifuged at 800× *g* and RT for 15 min in a 50 mL volume Leucosep tube containing 15 mL Ficoll-Paque Plus and 25 mL PBS. The mononuclear cell (PBMC) fraction containing the CTCs was carefully decanted in a new sterile 50 mL tube containing 1× CTL-Wash supplement (C.T.L., Cleveland, OH, USA) to maintain cell viability and functionality, and the Leucosep tube was washed with 5 mL PBS to prevent cell loss. Next, the total recovered volume was centrifuged for 10 min at 280× *g*, and the resulting cell pellet was used for CTC enrichment.

#### 4.5.1. Positive Enrichment of CTCs and the IsoFlux System

Positive enrichment of CTCs was based on cell surface EpCAM expression and was performed with the CTC Enrichment Kit and the IsoFlux system (Fluxion Biosciences), following the manufacturer’s instructions and as we have previously described [50]. The first tube of blood was discarded to avoid contamination with epithelial cells during sample extraction. The PBMC pellet obtained as described above was resuspended in 480 μL binding buffer (BB) and transferred from the 50 mL tube to a new microcentrifuge tube. To prevent cell loss, the 50 mL tube was washed with 400 μL BB using the same micropipette tip, and the volume was merged with the previous one (880 μL total volume). From this cell fraction, and prior to immunolabeling with anti-EpCAM-coated magnetic microbeads, nonspecific binding sites were blocked with 40 μL of blocking solution for 5 min on ice. Next, we added 40 μL of immunomagnetic beads, and cells were incubated in rotation for 2 h at 4 °C. After that, cells were transferred to a microfluidic cartridge previously primed with 3mL of BB, and the EpCAM^+^ cells were magnetically separated from the rest of the cells using the IsoFlux system. When the CTC enrichment protocol was complete, the cartridge was immediately removed from the IsoFlux and the magnetic beads were recovered from the CellSpot cap with 3 washes of 50 μL each of BB or stem cell culture medium. Recovered cells were immediately fixed for subsequent immunolabeling and CTC count or used to promote the growth of CSC spheroids in 6-well ULA plates, with normoxia and high-glucose DMEM/F12 medium supplemented as described in Section 4.4.

#### 4.5.2. Negative Enrichment of CTCs

Negative cell enrichment based on the expression of the leukocyte surface marker CD45 was alternatively used for the isolation of CTCs. The cell pellet obtained in Section 4.5 from 7 mL of peripheral blood was incubated with 200 µL 1× red blood cell lysis buffer for 10 min at RT. After centrifuging for 10 min at 300× *g*, cells were resuspended in 160 µL of rinsing solution (RS) with 0.5% BSA. Next, we added 40 µL of anti-CD45-coated magnetic microbeads and we incubated the sample for 15 min at 4 °C. Cells were then washed with 3 mL of RS and centrifuged at 300× *g* for 10 min. The resulting pellet was resuspended in 500 µL of RS and deposited on an LD column placed in a magnetic separator previously equilibrated with 2 mL of RS. While the sample was entering the column, we performed 3 washes with RS, which prevented the column from drying out, and collected the eluted fraction of CD45^−^ cells in 4 microcentrifuge tubes. Cells were harvested by centrifuging for 5 min at 300× *g* and finally cultured in 24-well ULA plates, with normoxia and/or hypoxia and high glucose DMEM/F12 medium supplemented as described in Section 4.4. During this protocol, each time the cell pellet had to be resuspended, half of the volume was used to disaggregate the cells and the other half to clean the pipette tip and prevent cell loss. All reagents were from Miltenyi Biotec (Bergisch Gladbach, Germany).

### 4.6. CTC Immunolabeling and Enumeration

In order to identify CTCs for enumeration, EpCAM^+^ cells resulting from the IsoFlux system (Section 4.5.1) were fixed with 1.85% formaldehyde for 20 min and subsequently immunolabeled with the CTC Enumeration Kit (Fluxion Biosciences), as we have previously described [50]. The time elapsed between fixation and cell labeling did not exceed 72 h. During this time, the fixed cells were kept in BB at 4 °C. To retain the immunomagnetically labeled cells and to not lose them during consecutive washing and incubation steps at RT, we used a small but powerful magnet. Fixed cells were first blocked with 10% normal donkey serum for 5 min, then incubated with 1:100 rabbit anti-human CD45 primary antibody for 20 min and with 1:200 donkey anti-rabbit IgG-Cy3 secondary antibody for 15 min in the dark. Before and after incubation with secondary antibodies, cells were washed with BB. After permeabilizing with 0.2% Triton X-100 for 5 min, cells were labeled with 1:50 mouse anti-human pan-cytokeratin-FITC in the dark for 50 min. Finally, the cell nuclei were stained with 1× Hoechst in 0.02% Triton X-100, and the sample was resuspended in a final volume of 40 µL BB before being mounted on a glass slide with mounting medium and the help of the magnet.

For the analysis, a single fluorescence image was obtained from the automatic scanning of each sample, using a spectral confocal microscope, LSM710 (Carl ZEISS, Oberkochen, Germany). Once the sample was delimited on the X and Y axes, we set at least four different Z planes to have a complete and unequivocal image that was analyzed with the ZEN Lite Blue/Black Edition free software (Carl ZEISS). CTCs were identified as CD45^−^ CK^+^ nucleated cells and CTC clusters were defined as a group of 3 or more CTCs aggregated in tight contact. We additionally counted double positive CD45^+^ CK^+^ cells. The cell counting was performed manually and independently by two trained members of the research group, who were blinded to the clinical data. Any disagreement was resolved by a third investigator.

### 4.7. Statistical Analysis

Statistical analyses were performed by using SPSS version 25.0 software (SPSS Inc. Chicago, IL, USA). Continuous variables were expressed as means ± standard deviations if they followed normal distribution or with medians and interquartile ranges if they had a skewed distribution. Categorical variables were expressed as frequencies (%). For comparisons involving a continuous variable, the Student’s *T*-test or the Mann–Whitney U test was used depending on the distribution of the variable. Correlation between continuous variables was determined by the Spearman test. Regarding categorical variables, the Chi-squared test was used, except for comparisons involving forecasted frequencies < 5, in which the Fisher test was used. The independent predictors of complete radiological response were assessed using multivariate logistic regression. ROC curves were derived from logistic regression, which allowed to calculate sensitivity, specificity, and positive/negative predictive values for complete radiological response. Cumulative tumor-free and overall survival were evaluated using Kaplan–Meier curves (long-rank test). All statistical analyses were two-tailed and a *p*-value of <0.05 was considered statistically significant.

## 5. Conclusions

The present pilot study suggested that the absence of clusters of CTCs at baseline is a predictor of a worse response to TACE in patients with HCC. Patients with large tumors without clusters of CTCs at baseline should not undergo TACE alone as a first line therapy, as complete radiological response is unlikely. Adjuvant systemic therapies should be tested in this subgroup of patients. Finally, the ex vivo culture of CTCs requires complex protocols which may include negative enrichment methods combined with a short period of hypoxia followed by normoxia. Future investigations should aim to prolong the duration of these cultures in order to study the properties of HCC-derived CTCs and their mutations conferring increased tumor aggressiveness and resistance to therapy.

## Figures and Tables

**Figure 1 ijms-24-02558-f001:**
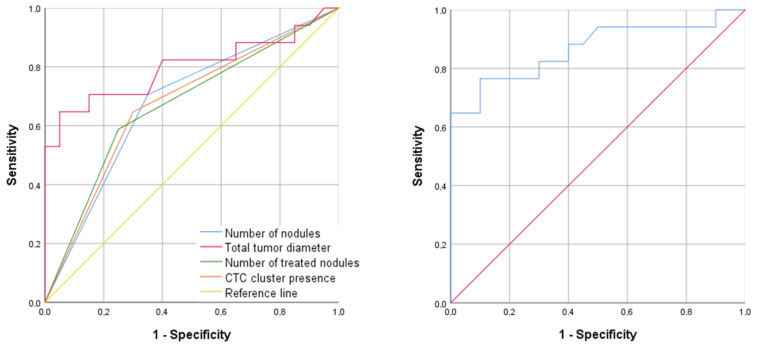
ROC curve showing the best diagnostic ability to predict patient response to TACE. (**Left** panel): individual ROC curves for the following predictors: multinodular disease, total tumor diameter, number of treated tumor nodules, and the presence of CTC clusters. (**Right** panel): ROC curve for the combination of total tumor diameter and CTC clusters at baseline by logistic regression.

**Figure 2 ijms-24-02558-f002:**
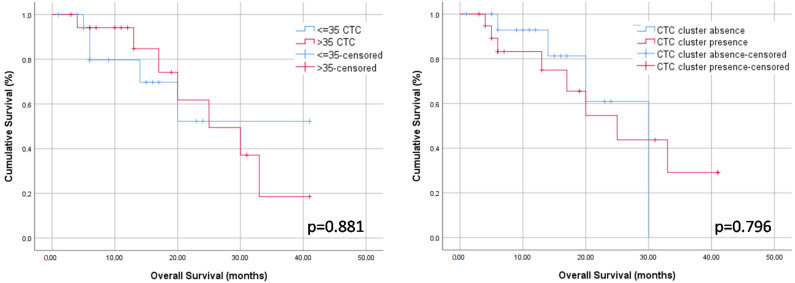
Kaplan–Meier curves (long-rank test) showing transplant-free survival rates according to baseline CTC count (**left** panel) and baseline presence of CTC clusters (**right** panel).

**Figure 3 ijms-24-02558-f003:**
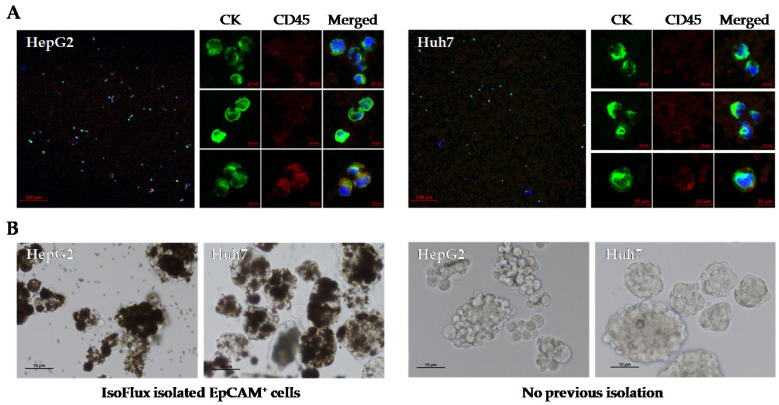
Utility of the IsoFlux system for obtaining cancer stem cell spheroids from established human HCC cell lines. EpCAM^+^ CTCs derived from cell lines HepG2 and HuH7 were isolated by the IsoFlux system and (**A**) visualized by immunofluorescence or (**B**) cultured under ultra-low adherence conditions to favor the growth of spheroids. Spheres from cells that were not previously isolated with magnetic beads are also shown. Growth on day 9 is shown.

**Figure 4 ijms-24-02558-f004:**
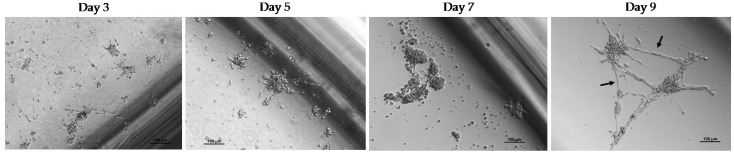
Follow-up of a culture of CTC spheroids derived from peripheral blood of a patient with HCC prior to TACE. CTCs were obtained by negative enrichment of CD45^−^ cells. Culture conditions were high glucose, normoxia, and ultra-low adherence. Images at 10×.

**Figure 5 ijms-24-02558-f005:**
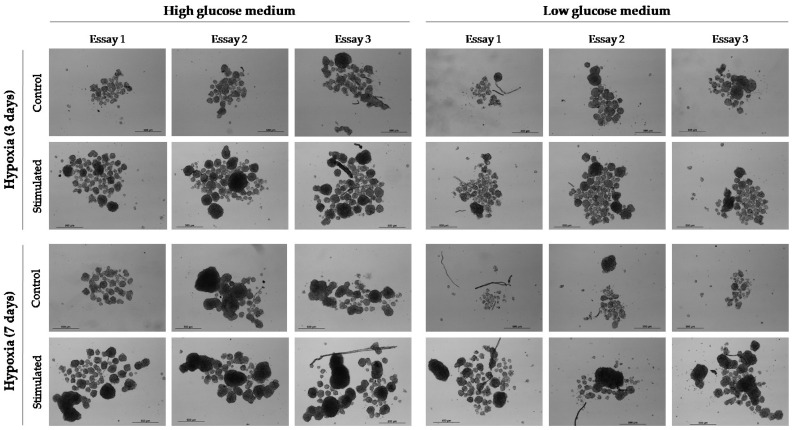
Effect of hypoxia (1% O_2_) and glucose deprivation (1 g/L (low glucose)) on the development and growth of spheroids from cell line HepG2. Cell culture conditions were maintained for 3 or 7 days. Thereafter, standard culture conditions (normoxia and 3.15 g/L glucose (high glucose)) were maintained. Images at 10× were taken 11 days after cell culture.

**Figure 6 ijms-24-02558-f006:**
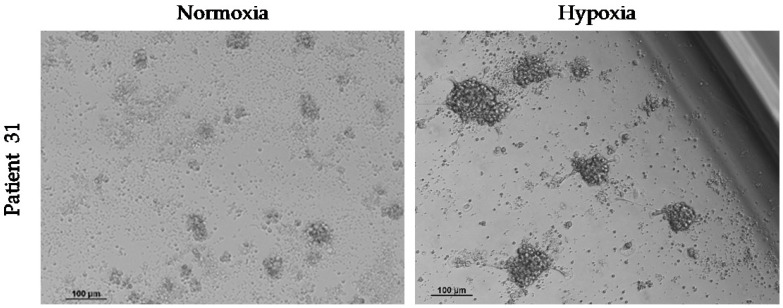
Effect of hypoxia on the growth of spheroids from CTCs in vitro. Hypoxia (1% O_2_) was maintained from the start of the cell culture of circulating CD45^−^ cells and for 3 days. Images at 10× were taken 4 days after culture.

**Table 1 ijms-24-02558-t001:** Clinicopathological characteristics of 37 prospectively enrolled patients with hepatocellular carcinoma undergoing transarterial chemoembolization as a first-line therapy.

Variable	
Age (years)	63.5 ± 7.9
Sex (male), n (%)	34 (91.9%)
Etiology	
-Hepatitis C (yes), n (%)	14 (37.8%)
-Hepatitis B (yes), n (%)	5 (13.5%)
-Alcohol (yes), n (%)	27 (73.0%)
-NAFLD (yes), n (%)	4 (10.8%)
Child–Pugh score, n (%)	
-A	32 (86.5%)
-B	5 (13.5%)
MELD score	9.0 (IQR 8.0–11.0)
BCLC stage, n (%)	
-0	1 (2.7%)
-A	25 (67.6%)
-B	8 (21.6%)
-C	3 (8.1%)
Portal hypertension (yes), n (%)	31 (83.8%)
Number of nodules, n (%)	
-Single nodule	18 (48.6%)
-Two nodules	11 (29.7%)
-Three nodules	3 (8.1%)
-Four nodules	5 (13.5%)
Main nodule diameter (cm)	3.0 (IQR 2.0–4.1)
Total tumor diameter (cm)	3.8 (IQR 2.9–5.7)
Number of treated nodules, n (%)	
-Single nodule	22 (59.5%)
-Two nodules	12 (32.4%)
-Four nodules	3 (8.1%)
AFP (ng/mL)	10.3 (IQR 3.7–49.0)
AST (U/L)	35.0 (IQR 25.5–50.0)
ALT (U/L)	26.0 (IQR 16.5–36.5)
GGT (U/L)	76.0 (IQR 46.5–171.0)
Tumor vascularity, n (%)	
-Hypervascularity	23 (62.2%)
-Moderate vascularity	8 (21.6%)
-Hypovascularity	6 (16.2%)

Abbreviations: IQR, interquartile range; NAFLD, non-alcoholic fatty liver disease; MELD, model for end-stage liver disease; BCLC, Barcelona clinic liver cancer staging system; AFP, alpha fetoprotein; AST, aspartate aminotransferase; ALT, alanine aminotransferase; GGT, gamma-glutamyl transferase.

**Table 2 ijms-24-02558-t002:** Relationship between clinicopathological characteristics and liquid biopsy findings, including individual count of circulating tumor cells and presence/absence of clusters of circulating tumor cells.

Variable	CTC Count (n = 37; 100.0%)			Cluster Presence (n = 20; 54.1%)	
	Patients n (%)	CD45^−^ CK^+^ Cells		Patients n (%)	*p*
		Median (IQR)	*p*		
Age (years)					
-<65	18 (48.6%)	36.5 (21.8–131.8)	0.504	10 (27.0%)	0.858
-≥65	19 (51.4%)	33.0 (14.0–56.0)		10 (27.0%)	
Sex (male)	34 (91.9%)	36.5 (21.8–95.0)	0.824	18 (48.6%)	1.000
Etiology					
-Hepatitis C (yes)	14(37.8%)	33.5 (19.8–216.5)	0.638	8 (21.6%)	0.769
-Hepatitis B (yes)	5 (13.5%)	49.0 (31.5–72.0)	0.790	3 (8.1%)	1.000
-Alcohol (yes)	27 (73.0%)	39.0 (21.0–110.0)	0.932	13 (35.1%)	0.288
-NAFLD (yes)	4 (10.8%)	36.0 (23.0–139.8)	0.941	3 (8.1%)	0.609
Child–Pugh score					
-A	32 (86.5%)	36.5 (21.3–88.3)	0.722	17 (45.9%)	1.000
-B	5 (13.5%)	24.0 (18.0–896.5)		3 (8.1%)	
MELD score					
-<9	14 (37.8%)	36.5 (20.5–62.8)	0.826	5 (13.5%)	0.081
-≥9	23 (62.2%)	33.0 (21.0–119.0)		15 (40.5%)	
BCLC stage					
-0-A	26 (70.3%)	33.5 (21.8–111.8)	0.842	12 (32.4%)	0.169
-B-C	11 (29.7%)	39.0 (14.0–90.0)		8 (21.6%)	
Portal hypertension (yes)	31 (83.8%)	34.0 (16.0–110.0)	0.650	16 (43.2%)	0.667
Number of nodules					
-Single nodule	18 (48.6%)	32.5 (21.8–112.3)	0.915	9 (24.3%)	0.630
-Multinodular	19 (51.4%)	39.0 (16.0–90.0)		11 (29.7%)	
Main nodule diameter (cm)					
-<3	17 (45.9%)	54.0 (27.0–157.5)	0.131	11(29.7%)	0.231
-≥3	20 (54.1%)	25.0 (17.5–54.3)		9 (24.3%)	
Total tumor diameter (cm)					
-<5	24 (64.9%)	41.5 (21.3–115.3)	0.656	14 (37.8%)	0.478
-≥5	13 (35.1%)	31.0 (19.5–72.0)		6 (16.2%)	
Number of treated nodules					
-Single nodule	22 (59.5%)	33.5 (22.8–89.8)	0.938	12 (32.4%)	0.942
->1 nodule	15 (40.5%)	49.0 (14.0–117.0)		8 (21.6%)	
AFP (ng/mL)					
-Normal (<9)	17 (45.9%)	49.0 (22.5–104.5)	0.498	8 (22.9%)	0.404
-Abnormal (≥9)	18 (48.6%)	33.0 (15.5–68.8)		11 (31.4%)	
-Missing	2 (5.4%)				
AST (U/L)					
-Normal (<35)	18 (48.6%)	44.0 (22.5–112.3)	0.670	11 (29.7%)	0.402
-Abnormal (≥35)	19 (51.4%)	33.0 (16.0–56.0)		9 (24.3%)	
ALT (U/L)					
-Normal (<27)	19 (51.4%)	31.0 (14.0–110.0)	0.637	13 (35.1%)	0.072
-Abnormal (≥27)	18 (48.6%)	41.5 (22.8–91.5)		7 (18.9%)	
GGT (U/L)					
-Normal (<75)	17 (45.9%)	33.0 (14.0–114.5)	0.532	10 (27.0%)	0.591
-Abnormal (≥75)	20 (54.1%)	51.0 (23.0–88.3)		10 (27.0%)	
Tumor vascularity					
-Hypervascularity	23 (62.2%)	49.0 (23.0–119.0)	0.026	13 (35.1%)	0.699
-Medium/hypo-vascularity	14 (37.8%)	23.5 (12.5–54.3)		7 (18.9%)	

**Table 3 ijms-24-02558-t003:** Univariate and multivariate logistic regression comparing patients with and without radiological response to TACE. Radiological response was evaluated one month after the procedure by dynamic imaging techniques following the mRECIST criteria, on the group of responders who attained a complete radiological response.

Variable	Univariate Analysis				Multivariate Analysis	
	Responder (n = 20; 54.1%)	Non-Responder (n = 17; 45.9%)	OR (95% CI)	*p*	OR (95% CI)	*p*
Age (years)	63.7 ±7.6	63.2±8.3	1.0 (0.9–1.1)	0.871	-	-
Etiology						
-Hepatitis C (yes), n (%)	8 (21.6%)	6 (16.2%)	0.8 (0.2–3.1)	0.769	-	-
-Hepatitis B (yes), n (%)	2 (5.4%)	3 (8.1%)	1.9 (0.3–13.2)	0.503	-	-
-Alcohol (yes), n (%)	13 (35.1%)	14 (37.8%)	2.5 (0.5–11.8)	0.244	-	-
-NAFLD (yes), n (%)	3 (8.10%)	1 (2.7%)	0.4 (0.0–3.8)	0.389	-	-
Child–Pugh score, n (%)						
-A	17 (45.9%)	15 (40.5%)	1 (Ref.)	0.775	-	-
-B	3 (8.1%)	2 (5.4%)	0.8 (0.1–5.1)		-	
MELD score	9.5 (IQR 7.3–11.0)	9.0 (IQR 8.0–11.0)	0.9 (0.7–1.2)	0.428	-	-
BCLC stage, n (%)						
-0-A	17 (45.9%)	9 (24.3%)	1 (Ref.)	0.041	-	-
-B-C	3 (8.1%)	8 (21.6%)	5.0 (1.1–23.8)		-	
Portal hypertension (yes), n (%)	18 (48.6%)	13 (35.1%)	0.4 (0.1–2.3)	0.278	-	-
Number of nodules, n (%)						
-Single nodule	13 (35.1%)	5 (13.5%)	1 (Ref.)	0.035	-	-
-Multinodular	7 (18.9%)	12 (32.4%)	4.5 (1.1–17.9)		-	
Main nodule diameter (cm)	2.5 (IQR 2.0–3.4)	3.6 (IQR 2.2–4.8)	1.8 (1.0–3.2)	0.055	-	-
Total tumor diameter (cm)	3.3 ± 1.0	5.8 ± 2.5	2.2 (1.3–3.9)	0.005	2.5 (1.3–4.8)	0.006
Number of treated nodules, n (%)						
-Single nodule	15 (40.5%)	7 (18.9%)	1 (Ref.)	0.041	-	-
->1 nodule	5 (13.5%)	10 (27.0%)	4.3 (1.1–17.4)		-	
AFP (ng/mL)	12.9 (IQR 4.9–49.0)	6.0 (IQR 3.1–46.0)	1.0 (1.0–1.0)	0.956	-	-
AST (U/L)	34.0 (IQR 25.0–45.5)	35.0 (IQR 26.5–63.5)	1.0 (1.0–1.0)	0.563	-	-
ALT (U/L)	28.0 (IQR 18.3–36.5)	20.0 (IQR 16.0–38.5)	1.0 (1.0–1.0)	0.457	-	-
GGT (U/L)	86.5 (IQR 45.0–145.0)	75.0 (IQR 51.5–177.5)	1.0 (1.0–1.0)	0.516	-	-
CD45^−^CK^+^ CTC count	49.0 (IQR 23.3–118.5)	26.0 (IQR 14.0–55.0)	1.0 (1.0–1.0)	0.648	-	-
CD45^+^CK^+^ cell count	30.0 (IQR 15.3–71.5)	26.0 (IQR 18.0–51.0)	1.0 (1.0–1.0)	0.386	-	-
Total cell count (CD45^−/+^CK^+^)	91.5 (IQR 45.3–154.8)	48.0 (IQR 35.5–122.5)	1.0 (1.0–1.0)	0.822	-	-
CTC clusters (yes), n (%)	14 (37.8%)	6 (16.2%)	0.2 (0.1–0.9)	0.039	0.2 (0.0–1.0)	0.049
Tumor vascularity, n (%)						
-Hypervascularity	12 (32.4%)	11 (29.7)	1 (Ref.)	0.769	-	-
-Medium/hypo-vascularity	8 (21.6%)	6 (16.2%)	0.8 (0.2–3.1)		-	

OR indicates odds ratio; CI indicates confidence interval.

**Table 4 ijms-24-02558-t004:** Area under receiver operating characteristic curve (AUROC), sensitivity (SN), specificity (SP), and positive (PPV) and negative (NPV) predictive values of the univariate and multivariate models in Figure 1.

Variable	AUROC (95% CI)	SN (95% CI)	SP (95% CI)	PPV (95% CI)	NPV (95% CI)
**Univariate analysis**					
Number of nodules	0.7 (0.5–0.9)	72.2 (48.9–95.7)	63.2 (38.8–87.5)	65.0(41.6–88.4)	70.6 (46.0–95.2)
Total tumor diameter	0.8 (0.6–1.0)	77.3 (57.5–97.1)	80.0 (56.4–100.0)	85.0 (66.9–100.0)	70.6 (46.0–95.2)
Number of treated nodules	0.7 (0.5–0.8)	68.2 (46.5–89.9)	66.7 (39.5–93.9)	75.0 (53.5–96.5)	58.8 (32.5–85.2)
CTC cluster presence	0.7 (0.5–0.9)	70.0 (74.4–92.6)	64.7 (39.1–90.4)	70.0 (47.4–92.6)	64.7 (39.1–90.4)
**Multivariate analysis**					
Combination of total tumor diameter and presence of CTC cluster	0.9 (0.7–1.0)	81.8 (63.4–100.0)	86.7 (66.1–100.0)	90.0 (74.4–100.0)	76.5 (53.4–99.6)

**Table 5 ijms-24-02558-t005:** Data for overall survival-rates at 6, 14, and 24 months after TACE therapy.

Overall Survival-Rates	6 Months	14 Months	24 Months
≤35 CTC (n = 19; 51.4%)	79.6%	69.7%	52.3%
>35 CTC (n = 18; 48.6%)	94.1%	84.7%	61.8%
CTC cluster absence (n = 17; 45.9%)	92.9%	81.3%	60.9%
CTC cluster presence (n = 20; 54.1)	83.2%	83.2%	54.6%

**Table 6 ijms-24-02558-t006:** Results of ex vivo culture of CTC spheroids from HCC patients undergoing TACE. The table shows the cell enrichment strategy, the cell culture conditions, and the presence of cellular spheres.

Patient Number	CSC Enrichment Method	Cell Culture Conditions	CSC Spheres
01–11	Positive enrichment (IsoFlux; EpCAM^+^ cells)	Normoxia	-
12–15	Negative enrichment (depletion of CD45^+^ cells)	Normoxia	-
16			√
17			-
18			√
19			-
20			√
21–23			-
24	Negative enrichment (depletion of CD45^+^ cells)	Normoxia and Hypoxia	-
25			√
26–30			-
31			√
32–33			-

## Data Availability

Raw data used for this study will be granted to any investigator upon reasonable request which should be directed to the corresponding author.

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
