# Peer review of "Enumeration and Characterization of Circulating Tumor Cells in Patients with Hepatocellular Carcinoma Undergoing Transarterial Chemoembolization"

_ijms, 2023, doi:10.3390/ijms24032558_

Round 1

Reviewer 1 Report

This paper illustrates characterization of circulating tumor cells (CTC) in patents with HCC undergoing TACE. Primary outcome was complete radiological response. 

Comments

1. The impact of the presence of CTC in peripheral blood needs to be clarified. The absence was a positive factor for complete radiological response. But the sentence in abstract and in conclusion seems the opposite.

2.  CTC could be culture in negative selection (CD45 as the marker for selection) method but not positive selection (EpCAM as the marker for selection) method.  What else markers do those CTC expressed in culture after negative selection method?

3. 5 of 22 patents had CTC in cultures. Do their HCC have cancer stem cells in histology?

Author Response

# To the Reviewer 1:

  1. Original comment: The impact of the presence of CTC in peripheral blood needs to be clarified. The absence was a positive factor for complete radiological response. But the sentence in abstract and in conclusion seems the opposite.

Response 1: We apologize for the misunderstanding. We would like to note that there was no relationship between baseline CTC count and radiological response to TACE. However, we found that the absence of clusters of CTCs, which are groups of CTCs attached with each other, was associated with worse response to TACE. As we acknowledged in the discussion this was an unexpected and intriguing finding. We hypothesized that these clusters or groups of CTCs bring the attention of the immune system more effectively than isolated CTCs, thus promoting a more intense immunological response against the tumor, although this may be confirmed in future studies.  

We edited the sentence in the abstract to make it mor informative. In addition, we have modified the conclusion as follows to clarify this issue: “The present pilot study suggested that the absence of clusters of CTCs at baseline is a predictor of worse response to TACE in patients with HCC”.

  1. Original comment: CTC could be culture in negative selection (CD45 as the marker for selection) method but not positive selection (EpCAM as the marker for selection) method. What else markers do those CTC expressed in culture after negative selection method?

Response 2: In this work, we have not analyzed the expression pattern of cell markers that could define the subpopulation of circulating cancer stem cells. The objective of this study was to correlate the presence of circulating CSC with the patient response to TACE therapy. We hypothesized that the number of cellular spheres obtained ex-vivo would be proportional to the presence of CSC in peripheral blood. Different methodologies for obtaining CTC spheres are based on different isolation and cell culture methods, which were included in references 7, 9, 10, 12, and 13.  We experienced difficulties to obtain CTC-derived spheroids and we tested different approaches that we hope will be useful for future researchers.

The question that the reviewer raises is very interesting. However, because we tried to obtain stable long-cultures of cCSC and cells rapidly died off, we could not obtain enough cell mass to analyze the expression of CSC markers (CD44, CD90, and others). We hope to be able to perform this analysis in the near future. We have now included the following text into the discussion section: “Obtaining a stable culture of CTCs was not possible as, the CTC spheroids died rapidly, precluding sufficient cell mass for further marker expression analyses”.                                               

  1. Original comment: 5 of 22 patents had CTC in cultures. Do their HCC have cancer stem cells in histology?

Response 3: This is a very pertinent comment. HCC is diagnosis relies on typical imaging features (on dynamic CT or MRI) and biopsy of the tumor is not required in most cases. Since this study was based on routine clinical practice, we had no tissue samples available to perform the analysis suggested by the reviewer. Another study focused on patients undergoing surgical resection or liver transplantation, in whom histology specimens are available, would be required to study the role of cancer stem cells within the tumor.

Reviewer 2 Report

This manuscript reported CTCs cluster could function as a biomarker for HCC prognosis with TACE, the topic is novelty, and the results could also support their conclusion, I recommend the acceptance before some minor revisions.

1. TACE is a common treatment for HCC patients, and some new technologies were developed to improve TACE efficiency, I recommend the authors included these progress in the introduction section, and some references could be available for author's revision (1. Science Bulletin, 2020, 65 (20):1685-1687. 2. Journal of Controlled Release, 2020, 323: 635-643. 3. Journal of Controlled Release. 2022;350:122-131. )

2. The style of Table 2 and 3 could be revised more clearly.

3. The statistical method of Figure 2 should be supplied, Cox or log-rank?

Author Response

# To the Reviewer 2:

  1. Original comment: TACE is a common treatment for HCC patients, and some new technologies were developed to improve TACE efficiency, I recommend the authors included these progresses in the introduction section, and some references could be available for author's revision (1. Science Bulletin, 2020, 65 (20):1685-1687. 2. Journal of Controlled Release, 2020, 323: 635-643. 3. Journal of Controlled Release. 2022;350:122-131. )

Response 1: We have edited the introduction to include this technology as proposed by the reviewer. In the new version, it can be read as follows: “In recent years, promising new technological advances have been developed, aimed at improving the stability of pharmaceutical agents and the clinical performance of TACE [15] [16]. Currently, radiological response one month after the procedure may be achieved in the majority of patients but…”.

  1. Original comment: The style of Table 2 and 3 could be revised more clearly.

Response 2: As suggested by the reviewer we have revised tables 2 and 3 for improved clarity. Table 2 showed too much information in the previous version. To solve this, we have moved cell count data from CD45+ CK+ cells and the sum of CTCs (single positive CD45- CK+ cells) and double positive cells to a new supplementary table (suppl table 1). Since additional staining for CD45 may not be specific of hematopoietic cells, we have kept the original text with the most relevant information about this cell population, and the pertinent text paragraph in discussion section. Regarding table 3, we have deleted the ‘initial model’ of the multivariate analysis, which is not relevant for the final results. We feel that the new version of these tables are equally informative and much easier to interpret. 

  1. Original comment: The statistical method of Figure 2 should be supplied, Cox or log-rank?

Response 3: We have now included the statistical method used (log-rank) in the heading of Figure 2.

Round 2

Reviewer 1 Report

Thank you for the responses. I think, in abstract, OR 0.2 for the variable "CTC clusters at baseline" should have a clear reference for clarification. 

Author Response

Original comment: I think, in abstract, OR 0.2 for the variable "CTC clusters at baseline" should have a clear reference for clarification.

Answer: We agree that without an appropriate reference in the abstract, results regarding clusters of CTCs are hard to understand. We have added the following sentence to the abstract: "All patients had detectable CTCs at baseline and 20 patients (54.1%) showed CTC aggregates or clusters in peripheral blood." We have also made some minor edits to the abstract to reduce the number of words, thus trying to meet the journal requirements. 

Finally, regarding the change in the english language assessment of the reviewer (previously classified as "minor spell check required" and now as "extensive editing required") we would like to note that the version we downloaded has many spaces missing and joined words, which were not present in the version we submitted. We have tried to correct most of these typos.